# The Cortical Silent Period in the Cricothyroid Muscle as a Neurophysiologic Feature for Dystonia Observation: E-Field-Navigated Transcranial Magnetic (TMS) Study

**DOI:** 10.3390/biomedicines11051373

**Published:** 2023-05-05

**Authors:** Ivan Konstantinović, Braco Bošković, Joško Šoda, Krešimir Dolić, Zoran Đogaš, Mirko Lapčić, Vlatko Ledenko, Toni Vrgoč, Maja Rogić Vidaković

**Affiliations:** 1Neurosurgery Division, University Hospital of Split, 21000 Split, Croatia; ivan.konstan@gmail.com (I.K.); mirkolapcic@gmail.com (M.L.); vlatko.ledenko@gmail.com (V.L.); 2Otorhinolaryngology Department, University Hospital of Split, 21000 Split, Croatia; bboskovic01@gmail.com; 3Signal Processing, Analysis, and Advanced Diagnostics Research and Education Laboratory (SPAADREL), Faculty of Maritime Studies, University of Split, 21000 Split, Croatia; jsoda@pfst.hr; 4Diagnostic and Interventional Radiology Department, University Hospital of Split, 21000 Split, Croatia; kdolic79@gmail.com; 5Medical Radiology, School of Medicine, University of Split, 21000 Split, Croatia; 6Split Sleep Medical Centre, University Hospital of Split, 21000 Split, Croatia; zdogas@mefst.hr; 7Laboratory for Human and Experimental Neurophysiology, Department of Neuroscience, University of Split School of Medicine, 21000 Split, Croatia; toni.vrgoc@mefst.hr

**Keywords:** cortical silent period, motor-evoked potentials, MEP, transcranial magnetic stimulation, TMS, navigated TMS, dystonia, spasmodic dysphonia, laryngeal dystonia

## Abstract

The cortical silent period (cSP) is a period of electrical silence following a motor-evoked potential (MEP) in the electromyographic signal recorded from a muscle. The MEP can be elicited by transcranial magnetic stimulation (TMS) over the primary motor cortex site corresponding with the muscle. The cSP reflects the intracortical inhibitory process mediated by GABA_A_ and GABA_B_ receptors. The study aimed to investigate the cSP in the cricothyroid (CT) muscle after applying e-field-navigated TMS over the laryngeal motor cortex (LMC) in healthy subjects. Then, a cSP as a neurophysiologic feature for laryngeal dystonia was observed. We applied a single-pulse e-field-navigated TMS to the LMC over both hemispheres with hook-wire electrodes positioned in the CT muscle in nineteen healthy participants, which triggered the elicitation of contralateral and ipsilateral corticobulbar MEPs. The subjects were engaged in a vocalization task, and then we assessed the following metrics: LMC intensity, peak-to-peak MEP amplitude in the CT muscle, and cSP duration. The results showed that the cSP duration from the contralateral CT muscle was distributed from 40 ms to 60.83 ms, and from the ipsilateral CT muscle, from 40 ms to 65.58 ms. Also, no significant difference was found between the contralateral and ipsilateral cSP duration (t(30) = 0.85, *p* = 0.40), MEP amplitude in the CT muscle (t(30) = 0.91, *p* = 0.36), and LMC intensity (t(30) = 1.20, *p* = 0.23). To conclude, the applied research protocol showed the feasibility of recording LMC corticobulbar MEPs and observing the cSP during vocalization in healthy participants. Furthermore, an understanding of neurophysiologic cSP features can be used to study the pathophysiology of neurological disorders that affect laryngeal muscles, such as laryngeal dystonia.

## 1. Introduction

The primary motor cortex for laryngeal motor cortex (LMC) plays an important role in the human voice and speech production. The functional integrity of the LMC and its connections with cortical and subcortical brain regions warrants additional investigations due to scarce findings on neurophysiological mechanisms and unsuccessful treatments of disorders that affect laryngeal muscles such as laryngeal dystonia disease [1,2,3]. To date, the methodologies for mapping LMCs with transcranial magnetic stimulation (TMS) [4,5,6,7,8] and intraoperatively by electrical stimulation (ES) techniques [7,8,9,10] have been previously developed to record corticobulbar motor-evoked potentials (MEPs) from laryngeal muscles. Rogić Vidaković et al. [8], Deletis et al. [10], and Espadaler et al. [6] used an e-field-navigated transcranial magnetic stimulation to map the LMC for representation of the cricothyroid (CT) muscle with MEP recordings from the CT muscle. They reported a cortical distance of 25.19 ± 6.51 mm from the upper extremity muscle representation (abductor pollicis brevis (APB) muscle) to a laterally positioned LMC for CT muscle representation. Further, stimulating the LMC for a CT muscle representation triggered MEPs with mean latencies of 11.89 ± 1.26 ms/11.75 ± 2.07 ms (contralateral muscle) [6,8] and 11.75 ± 1.98 ms (ipsilateral muscle) [8]. Recently, Chen et al. [11] used line-navigated TMS to map the thyroarytenoid (TA) muscle representation by recording the MEPs from TA muscles in healthy subjects. Likewise, the stimulation over the LMC of the TA muscle elicited contralateral MEPs with a mean latency of 13.1 ± 2.0–2.3 ms and ipsilateral MEPs with a mean latency of 15.5–15.6 ± 2.3–2.8 ms.

Additional to corticobulbar MEP signal estimation, the measurement of the cortical silent period (cSP) was performed. The cSP is defined as the period of electrical silence recorded in the muscle immediately after the MEP response during subject engagement in a vocalization (phonation) task. By using focal TMS over the LMC with a suprathreshold intensity, a cSP can be evoked from contralateral and ipsilateral laryngeal muscles. It is generally accepted that the cSP reflects an intracortical inhibitory process mediated by GABA_A_ and GABA_B_ receptors [12,13]. Previous TMS studies indicated a reduced inhibition as a characteristic of hand dystonia, cervical dystonia, and focal laryngeal dystonia (spasmodic dysphonia) [14]. To date, a cSP was reported only for TA muscles as a measure of LMC excitability in healthy subjects [11] and patients with laryngeal dystonia [14] using a line-navigated TMS system [15,16]. The cSP duration for TA muscles ranged from 41.7 ms to 66.4 ms in healthy subjects [11], and was found to be shortened in laryngeal dystonia subjects [14]. Moreover, the excitability of the LMC representation for CT muscles—as investigated by the cSP measure of the GABA receptor-mediated inhibition process—has not been previously evaluated. In the context of applying the TMS methodology in the research of the corticospinal tract and corticobulbar tract pathophysiology of disorders that affect the laryngeal muscles, the methodological use of e-field-navigated TMS is recommended over line-navigated TMS [6,7,8,15,16]. Line-navigated TMS is susceptible to errors because there is no visualization of a spot of maximal stimulation if there is a slight coil tilt. On the contrary, e-field-navigated TMS computes the e-field maximum, where the cortex is best stimulated considering the head geometry, magnetic coil shape, orientation, location, and orientation of the cortical folds [15].

Therefore, the current study aimed to investigate the cSP in CT muscles after applying e-field-navigated TMS in healthy subjects to use the cSP as a neurophysiologic feature for laryngeal dystonia observations. The rest of the paper is organized as follows: Section 2 describes the experiment protocol, number of participants, procedure for electrode positioning, skin preparation, hand and laryngeal region assessments, and procedure for the e-field transcranial magnetic stimulation. The section also describes the measures and statistical tests used. In Section 3 the results are shown and described. Then, Section 4 describes the results’ implications and limitations, and future directions for potential studies. Finally, Section 5 gives conclusions about the proposed study.

## 2. Materials and Methods

### 2.1. Participants

The study enrolled 19 healthy participants with a mean age of 35.63 ± 8.08 years and an age range from 24 to 53 years. Further, 47.36% of participants were female and 89.47% were right-handed, with a normal mean body mass index of 24.03 kg/m^2^.

### 2.2. Procedure for Electrode Positioning

Participants were comfortably seated in an electronically adjustable chair with a head tracker attached to their forehead as a part of the frameless stereotactic e-field-navigated TMS system (Nexstim NBS System 4 of the manufacturer Nexstim Plc., Helsinki, Finland). The mapping of the primary motor cortex (M1) for hand muscle representation was assessed before mapping the LMC.

#### 2.2.1. Skin Preparation

The skin on the hand was cleaned with acetone with calcium chloride, and the surface layer of the skin was gently cleaned with fine sandpaper. The skin around the laryngeal prominence was cleaned with octenidine dihydrochloride and 2-phenoxyethanol (Octenisept, Schülke & Mayr). Lidocaine hydrochloride (Dolokain; 20 md/g gel), a local anesthetic from the group of amide compounds, was applied to the cleaned area.

#### 2.2.2. Hand Region Assessment

The MEPs were recorded from the hand APB muscle with a pair of self-adhesive surface electrodes (Ambu^®^ Blue Sensor BR, BR-50-K/12 of manufacturer Ambu A/S) in a belly-tendon montage. The electrodes were attached to the electrode cable of a Nexstim electromyography (EMG) apparatus with a 1.5 mm touch-proof female safety connector (DIN 42-802) coupled to a 6-channel EMG and common ground EMG amplifier (external module) with TMS-artefact rejection circuitry. EMG was an integrated part of the e-field-navigated TMS device.

#### 2.2.3. Laryngeal Region Assessment

Disposable paired hook-wire electrodes (IOM electrodes) (SGM d.o.o., Split, Croatia) were used to record corticobulbar MEPs and the cSP from CT muscles. After the electrode wire was inserted into the CT muscle, the hypodermic needle was withdrawn and the flexible electrode wire was securely fixed in the muscle. We used the insertion needles of 50 mm in length with a caliber of 7 mm (22 G), a lead wire length of 0.40 mm, and 2 PTFE-insulated stainless steel wires with a blue touch-proof connector. The insertion method of the hook-wire electrodes into the CT muscle was similar to those reported in our previous studies [6,8,10]. Anatomical landmarks were first drawn on the patient’s skin. The insertion point was marked on the level of the cricothyroid notch, five millimeters lateral to the midline. The needle was inserted with a movement tangential to the cricothyroid membrane (app. 50°) until the electrode passed into the muscle. To confirm the correct position of the electrode, the participants were asked to phonate the sound /i/ at a low pitch and slightly raise the pitch. With the correct electrode positioning in the CT muscle, the EMG activity suddenly increased and the interference patterns became dense. The movement of the sternohyoid muscle was verified by raising the participant’s head or pressing the fingers against his/her forehead. In the case of EMG activity, the needle was reinserted with these actions (infrahyoid muscle placement). During insertion, the electrodes were connected to an EMG cable to allow the monitoring of muscle activity in real time. An experienced otolaryngologist inserted the paired hook-wire electrodes in the CT muscle (BB; author of the study). In four participants, paired hook-wire electrodes were inserted in the right CT muscle and in fifteen participants, in the left CT muscle. Contralateral and ipsilateral MEPs and cSPs were collected from thirteen participants, contralateral MEPs and cSPs from five participants, and ipsilateral MEPs and cSPs were collected from one subject.

### 2.3. E-Field-Navigated Transcranial Magnetic Stimulator

An anatomical T1 magnetic resonance image (MRI) was performed at least 24 h before the TMS experiment. The brain MRI was performed using a Siemens Magnetom Avanto 1.5 T device (Siemens Healthcare GmbH). The MRI images were 3D-reconstructed according to the individual brain anatomy (3D optical tracking unit of Polaris^®^ Vicra) [17]. A figure of eight-shaped magnetic coil was used, generating a biphasic pulse of 289 µs in length. The eight-shaped coil, with an inner winding diameter of 50 mm and an outer winding diameter of 70 mm, was tangentially placed in a posterior-to-inferior direction to the subject’s skull over the primary motor cortex. The maximal electric field was 172 V/m below the focal coil of the e-field-navigated TMS in the spherical conductor model representing the human head. The MRI image was imported into the e-field-navigated TMS system (Nexstim NBS System 4 of the manufacturer Nexstim Plc., Helsinki, Finland) to guide the localization of the LMC in the primary motor cortex.

Hand region excitability/primary motor cortex for hand muscle representation (M1) was evaluated using surface electrodes attached to the APB muscle to determine the resting motor threshold (RMT). The RMT was defined as the lowest stimulation intensity used to elicit a minimum of 5 positive MEP responses out of 10 trials that had peak-to-peak amplitudes higher than >50 µV [18].

Laryngeal motor cortex (LMC) excitability was evaluated by following the guidelines of previously published protocols for mapping the LMC whilst recording MEPs from CT muscles [6,8]. After obtaining RMTs for the APB muscle, the coil was tangentially and laterally moved to the central sulcus over the LMC representation, approximately 25.19 ± 6.51 mm laterally from the APB hot-spot [6]. The cSP intensity was defined as the TMS intensity that elicited a cSP in five out of ten consecutive trials. Single-pulse cortical stimulations were performed during the high-pitch vocalization of the sound /i/. Participants were instructed to produce a comfortable pitch during the vocalization task. The single pulse was applied approximately one to two seconds after the initiation of the continuous vocalization. The MEPs and cSPs were collected over the contralateral and ipsilateral hemispheres. The stimulation intensity for detecting the cSPs from CT muscles was approximately 130%–150% of the RMTs for the hand muscle representation.

### 2.4. Outcome Measures and Data Processing

The outcome measures included the RMT intensity for the APB muscle, MEP latency, and MEP amplitude for the MEP response elicited from the APB muscle at the RMT intensity. The outcome measures from the CT muscle included the intensity of stimulation (expressed as a percentage of the maximal stimulator output) required to elicit the cSP from the CT muscle, the duration of the cSP, and the amplitude of MEPs elicited from the CT muscle preceding the cSP. As the cSP was detected in the CT muscle during vocalization and, therefore, pre-EMG activation preceded the MEP response, the latency of the MEP response from the CT muscle was not estimated. The MEP latency from the CT muscle was previously reported by applying different stimulation protocols and designs [6,7,8].

The beginning of the cSP was defined as the timepoint of the magnetic pulse (zero time in Figure 1). Therefore, the cSP duration was calculated by subtracting the onset from the offset of the cSP (Figure 1) [11]. The MEP amplitude and latency from the APB, the MEP amplitude from the CT muscle, and the cSP were estimated by a custom-made Matlab script (R2021a) [19]. The MEP response from the CT muscle was detected within a time window range of 10–30 ms after the stimulus artifact, whilst the MEP amplitude of the APB muscle was identified within the range of 18–30 ms [19]. An automatic algorithm calculated the MEP response from the CT muscle, whilst the cSP duration offset was set manually [19]. Five to ten trials were minimally taken for the MEP and cSP analyses.

### 2.5. Data Analysis

The parametric statistic was used because skewness and kurtosis parameters did not indicate great deviations from a normal distribution. Participants’ characteristics were presented using descriptive statistics. Also, previous studies confirmed no significant differences in MEP latency and cSP duration in TA muscles concerning left and right hemispheric stimulation [11]. Further, as the left hemisphere stimulation was stimulated in the majority of participants (fifteen) in our study, the data on cSP duration and MEP amplitude in CT muscles were presented as contralateral and ipsilateral responses in the present study. Groups were compared with Student *t*-tests for independent samples. The equality of variances between groups was calculated by Levene’s test. A *p* < 0.05 was considered statistically significant. Statistica 12 software was used for the data analysis.

## 3. Results

Table 1 presents the TMS measures for upper extremity APB mapping (RMT intensity, MEP latency, and MEP amplitude) for right and left hemisphere stimulation during the resting condition. Further, TMS measures (cSP duration, MEP amplitude, and LMC intensity) for mapping the LMC for the laryngeal CT muscle were represented during vocalization for contralateral and ipsilateral LMC hemispheric stimulations. We observed and calculated no significant difference for RMT intensity (t(31) = 0.98, *p* = 0.33) between the left and right hemisphere stimulation for the APB muscle representation, MEP latency (t(31) = 0.94, *p* = 0.35), and MEP amplitude (t(31) = 0.86, *p* = 0.39) between the left and right APB muscle (Table 1). Furthermore, no significant difference was estimated between the contralateral and ipsilateral cSP duration (t(30) = 0.85, *p* = 0.40), MEP amplitude in the CT muscle (t(30) = 0.91, *p* = 0.36), and LMC intensity (t(30) = 1.20, *p* = 0.23) (Table 1). The contralateral cSP duration range in the CT muscle was 40 to 60.8 ms and the ipsilateral cSP duration range was 40 to 65.6 ms (Figure 2). Figure 3 depicts the repeatability of the cSP in the CT muscle in a single subject. Moreover, no significant difference was found for contralateral cSP duration between the left and right CT muscle (t(16) = −0.93, *p* = 0.36).

## 4. Discussion

The study evaluated the corticobulbar excitability of the LMC using a cSP measure with hook-wire electrodes positioned in the CT muscles in healthy subjects. The MEP responses elicited by the navigated TMS were confirmed as having a cortical origin due to the bilateral responses following magnetic cortical stimulation [6,7,8,10,11]. The study results suggested that the cSP may be used as a neurophysiologic measure of corticobulbar excitability for the LMC for CT representation additional to MEP latency and MEP amplitude measures.

The cSP was initially revealed by Merton and Morton in 1980 as the electrical silence measure recorded in an EMG signal following an MEP response whilst electrical or magnetic stimulation was delivered to the M1 during tonic muscle contraction [18,19,20]. Also, previous reports suggest that both the spinal and cortical mechanisms may justify the duration of the cSP as well as subcortical and cortical–subcortical interplays [21], but the general agreement is a predominant role of intracortical inhibitory phenomena in the genesis of the cSP [18,22]. By mapping the M1 for hand muscle representation using a suprathreshold intensity, the cSP can be evoked in a contralateral hand muscle lasting up to 100–300 ms following an MEP, with the level of contraction not interfering with the cSP duration [23]. Furthermore, studies investigating the cSP by mapping the M1 for facial and tongue muscle representation reported a cSP duration between 69–169 ms in the triangularis, 68–111 ms in the orbicularis oculi, and 64.2 ± 4.5 ms in the tongue [24,25,26]. The recent work by Chen et al. [11] provided evidence on cSP duration in laryngeal TA muscles, ranging from 41.7 to 66.4 ms in healthy subjects. Also, Chen et al. [14] reported findings on significantly shortened cSP durations in TA muscles in patients with laryngeal dystonia. The duration of contralateral and ipsilateral cSPs for the laryngeal CT muscles reported in the proposed work was of a similar duration for the cSPs in TA muscles, as reported by Chen et al. [11]. The cSP duration values collected from the left and right TA muscles, regardless of hemisphere stimulation, were similar [11], and the data of our work supported these findings for cSPs in CT muscles. It has to be emphasized that in the present work, an analysis of the ipsilateral cSP duration between the left and right CT muscles, regardless of hemispheric stimulation, was not reported because an ipsilateral cSP was recorded from the right CT muscle in one subject.

The MEP amplitude preceding the cSP was not observed to modulate with changes in TMS intensity during the vocalization task, similar to the findings of Chen et al. [14]. Further, when mapping the LMC for CT muscles with e-field-navigated TMS in healthy subjects (during resting conditions and with transcranial electrical stimulation (TES) in patients during general anesthesia), the MEP amplitudes proved to be significantly higher in contralateral MEP responses compared with ipsilateral MEP responses. The significantly higher MEP amplitudes in the contralateral CT muscles recorded during the resting condition (no vocalization involved) were explained by the CT muscle’s bilateral nature and the contralateral projections’ predominance [8]. However, when activating the CT muscle by voluntary tonic activation, such as during the vocalization (phonation) engagement in the assessment of the cSP, no MEP amplitude differences between the contralateral and ipsilateral MEP responses preceding the cSP were observed.

A potential limiting factor might be related to the fact that hook-wire electrodes were not inserted in both the left and right CT muscles in the whole sample of participants. However, due to previous findings of no reported significant differences for cSP duration in the left and right intrinsic laryngeal muscles [11,14], regardless of hemisphere stimulation, we believed that the limiting factor was rather negligible. Furthermore, the present study included healthy subjects with a mean age of 35 ± 8.08 years and a large age range (24–53 years), which might be considered to be a limiting factor. In contrast, Chen et al. [11] included a smaller number of healthy subjects (N = 11) compared with our study (N = 19), with a mean age of 54 ± 7.4 years, but without stating the age range. In another study of Chen et al. [14], 16 healthy subjects were included with a mean age of 51.5 ± 7.7 years, but the age range information was similarly not elaborated. Therefore, it was impossible to discuss the possible age differences in the context of cSP investigations in CT muscles in healthy subjects.

We want to point out that an advantage of the present study was the use of e-field-navigated TMS in contrast to Chen et al. [11] and Chen et al. [14], who used line-navigated TMS. The reason for recommending e-field-navigated TMS was to reduce technical errors, mainly when the magnetic coil is not continuously and tangentially positioned against the subject’s head [27]. The maximal activation is supposed to be located on the line that passes through the coil center, perpendicular to the coil’s bottom surface [15]. This means that a slight tilting of the coil would render errors in estimating the supposed maximal stimulation. Sollmann et al. [28] reported on differences in clinical applicability, workflow, and M1 mapping results between e-field-navigated TMS and line-navigated TMS in patients with brain tumors. The ratio of positive motor spots was significantly higher for e-field-navigated TMS compared with line-navigated TMS. Also, the cortical distances of the motor hot-spots between e-field-navigated TMS and line-navigated TMS were 8.3 ± 4.4 mm on the ipsilateral hemisphere and 8.6 ± 4.5 mm on the contralesional hemisphere. The advantages of e-field-navigated TMS were a shorter time per stimulation, a higher rate of positive responses, and an increase in mapping speed [28]. Sollmann et al. [28] also reported a different spread of spatial motor maps for upper and lower extremity muscles with a partial overlapping of motor maps when comparing e-field-navigated TMS and line-navigated TMS. Therefore, the motor mapping results between e-field-navigated TMS and line-navigated TMS shown by Sollmann et al. [28] should be considered in further investigations of corticobulbar excitability and pathophysiology in disorders that affect laryngeal muscles. Future studies might compare cSP measurements on a larger sample of healthy subjects, covering wider age variations with both techniques (e-field-navigated TMS and line-navigated TMS) to determine the cSP reference values in patients with diseases affecting laryngeal muscles.

## 5. Conclusions

The present study investigated the corticobulbar excitability of the LMC using e-field-navigated TMS-evoked cSPs in the CT muscles in healthy subjects. The cSP duration parameters were in line with the parameters of the TA muscles [11,14]. Given there are reports of significantly shortened durations of the cSP in the TA muscle in adductor laryngeal dystonia patients [14], the measurements of the cSP in the CT muscle represent an adjuvant neurophysiologic feature for the investigation of intracortical inhibitory phenomena in disease conditions such as laryngeal dystonia. It is due to the fact that the TA and CT muscles are innervated by different branches of the cranial vagal nerve, the recurrent laryngeal nerve branches innervating the TA muscle, and the superior laryngeal branch innervating the CT muscle.

It could be concluded that the e-field-navigated TMS assessment of LMC corticobulbar excitability with cSP measurements in CT muscles was successfully performed in healthy subjects. The use of hook-wire electrodes to measure the cSP of the LMC provided an additional tool to evaluate the GABA receptor-mediated inhibition process in the LMC for CT muscle representation. Therefore, the cSP from CT muscles can be applied in future studies as a neurophysiologic feature for laryngeal muscle diseases such as laryngeal dystonia.

## Figures and Tables

**Figure 1 biomedicines-11-01373-f001:**
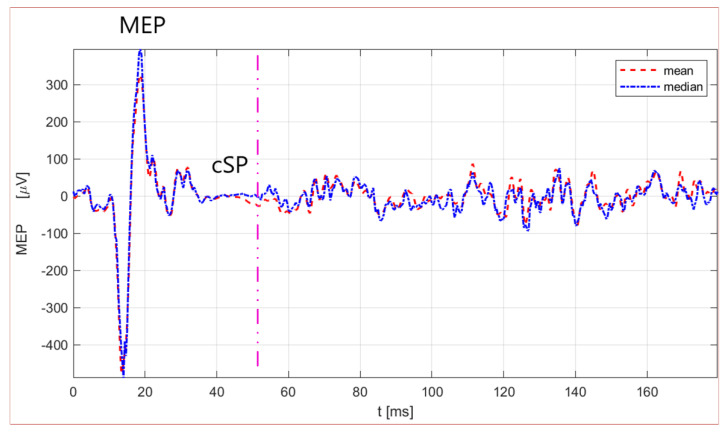
MEP responses and detection of cSP from laryngeal CT muscle. Six individual trials are presented as mean (red dashed line) and median (blue dashed line) for a male subject (No. 9). The pink vertical dashed line represents the offset of the cSP response, whilst the magnetic stimulation onset is represented as the zero time. The time on the x-axis is expressed in milliseconds (ms), and on the y-axis is the MEP amplitude in microvolts (µV).

**Figure 2 biomedicines-11-01373-f002:**
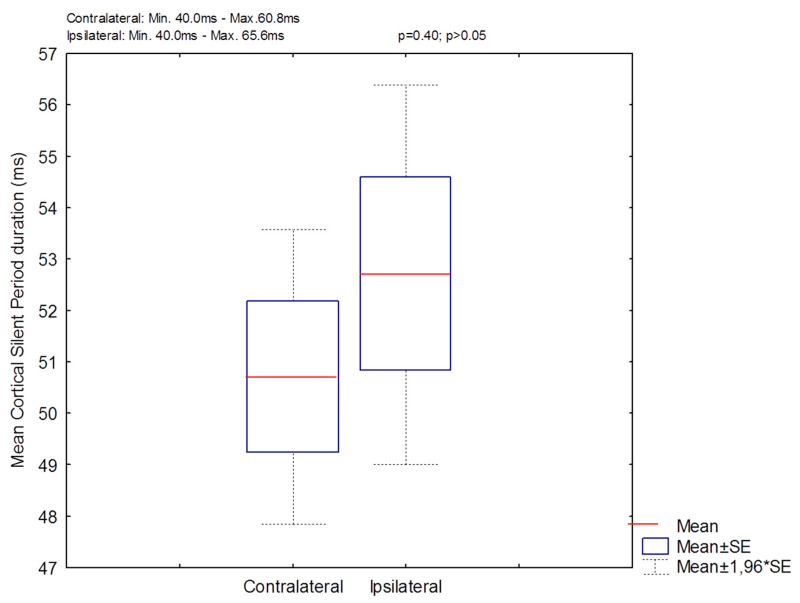
Graphical representation of cSP duration for contralateral and ipsilateral responses. The central red line specifies the median and mean value, and the bottom and top edges of the box show the 25th and 75th percentiles. The whiskers represent mean ± 1.96*SE of data. *p* > 0.05. SE: standard error of the mean.

**Figure 3 biomedicines-11-01373-f003:**
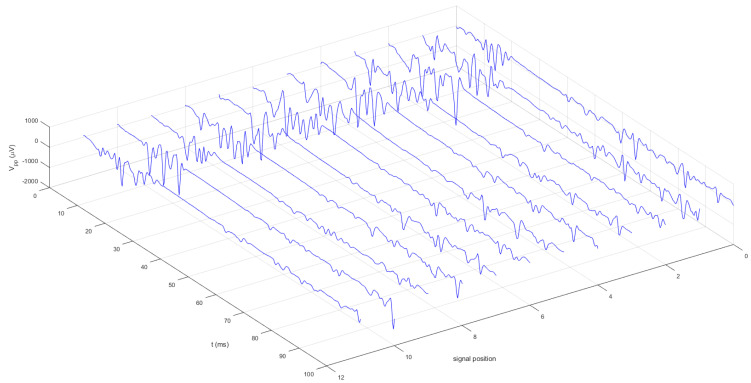
The 3D graphical representation of the repeatability of cSP in CT muscles in a single subject (No. 13). On the x-axis, the time in milliseconds is shown and on the y-axis, “Vpp” (peak-to-peak amplitude) denotes the MEP amplitude value expressed in microvolts. A total of twelve single trials are presented with MEPs in CT muscles followed by cSP. Pre-EMG activity is recorded preceding MEPs and following cSP.

**Table 1 biomedicines-11-01373-t001:** The differences in motor cortical mapping for upper extremity and laryngeal CT muscles.

		Mean ± SD (ms)	*t*-Test	*p*-Value
RMT Intensity for APB (%)	Right hemisphere	33.66 ± 5.40	*t* = 0.98	0.33
	Left hemisphere	35.83 ± 6.91		
MEP Latency APB (ms)	Right hand	23.53 ± 1.71	*t* = 0.94	0.35
	Left hand	22.91 ± 2.10		
MEP Amplitude APB (μV)	Right hand	298.05 ± 221.83	*t* = 0.86	0.39
	Left hand	238.93 ± 155.88		
cSP Duration CT (ms)	Contralateral	50.71 ± 6.19	*t* = −0.85	0.40
	Ipsilateral	52.70 ± 7.04		
MEP Amplitude CT (μV)	Contralateral	972.70 ± 656.47	*t* = −0.91	0.36
	Ipsilateral	1234.34 ± 961.88		
LMC Intensity (%)	Contralateral	55.76 ± 10.62	*t* = 1.20	0.23
	Ipsilateral	51.50 ± 8.83		

SD: standard deviation; RMT: resting motor threshold; APB: abductor pollicis brevis; cSP: cortical silent period; CT: cricothyroid muscle; LMC: laryngeal motor cortex; µV: microvolt; ms: milliseconds; %: percentage.

## Data Availability

All data generated during this study may be available on request.

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
