# Peer review of "The Cortical Silent Period in the Cricothyroid Muscle as a Neurophysiologic Feature for Dystonia Observation: E-Field-Navigated Transcranial Magnetic (TMS) Study"

_biomedicines, 2023, doi:10.3390/biomedicines11051373_

Round 1

Reviewer 1 Report

A silent period, known as the cortical silent period, cSP, can be observed following a motor-evoked potential (MEP) elicited by transcranial magnetic stimulation (TMS) over the primary motor cortex. The cSP believed to reflect an inhibitory process mediated by cortical GABAergic transmission.  Prior investigation by Chen et al indicates that the cSP recorded in the thyroarytenoid (vocal) muscle is shortened in patients with spasmodic dysphonia, suggesting that investigation of cSP in laryngeal muscles might be a useful procedure for investigation of cortical inhibitory processes in cases of laryngeal dystonia.

In a previous study the authors of the current manuscript had demonstrated the feasibility of employing neuronavigated TMS (nTMS) to identify the laryngeal motor cortex (LMC) site at which stimulation elicits MEPs in the cricothyroid muscle. In this manuscript, they report an investigation of the cSP following MEPs elicited by applying nTMS to the relevant LMC site during tonic muscle contraction maintained by vocalization, in 19 healthy participants. They demonstrated the feasibility of recording LMC intensity, corticobulbar MEPs and cSP in the cricothyroid muscle during vocalization in healthy participants,  They found no significant difference between contralateral and ipsilateral cSP duration, MEP amplitude in CT muscle, and intensity. These observations demonstrate the potential value of  nTMS together with EMG recordings from cricothyroid muscle for investigating the pathophysiology of disorders that affect laryngeal muscles.  

The methodology appears sound. The demonstration of the feasibility of recording the cSP from the cricothyroid muscle is a of potential value for the investigation of laryngeal muscle disorders

One minor issue is that fact that reference numbers are missing from the reference list, making the identification of references difficult.

Minor editing of English language is required. In particular, the first sentence of the Abstract is unclear, due to inappropriate placement of the phrase ‘following motor-evoked potential (MEP)’. 

I suggest:

The cortical Silent Period (cSP)  is a period of electrical silence following a motor-evoked potential (MEP) in the electromyographic signal recorded from a muscle. The MEP can be elicited by transcranial magnetic stimulation (TMS) over the primary motor cortex site corresponding to the muscle.

Author Response

The response to Reviewer 1 comments ara attached in a PDF document.

Reviewer 2 Report

The aims of this study was to investigate cSP in CT muscle after applying
nTMS in healthy subjects to use cSP as a neurophysiologic feature for laryngeal dystonia observation.

The study is very interesting, topical and well structured. I have only a few minor suggestions for authors. In particular, I believe that the "introduction" and "discussion" sections are too brief. In the introduction, it would be appropriate to better explain the importance of "nTMS" and why it is better than other methods. In view of the rather large age range (24-53) I think it is appropriate to mention this aspect within the limits of the study. Furthermore, paragraph 2.1 is a bit confusing and should be rewritten. As far as the "Discussion" paragraph is concerned, the results obtained and the contribution they can give to the scientific community should be better discussed.

Author Response

The author's response to Reviewer 2 comments is attached in a PDF document.
